# Cell Therapy for Parkinson’s Disease

**DOI:** 10.3390/pharmaceutics15122656

**Published:** 2023-11-22

**Authors:** Surabhi Shastry, Junkai Hu, Mingyao Ying, Xiaobo Mao

**Affiliations:** 1Neuroregeneration and Stem Cell Programs, Institute for Cell Engineering, Johns Hopkins University School of Medicine, Baltimore, MD 21205, USA; sshastr6@jh.edu (S.S.); jhu89@jhu.edu (J.H.); 2Department of Neurology, Johns Hopkins University School of Medicine, Baltimore, MD 21205, USA; 3Hugo W. Moser Research Institute at Kennedy Krieger, Baltimore, MD 21205, USA; 4Institute for NanoBioTechnology, Johns Hopkins University, Baltimore, MD 21218, USA; 5Department of Materials Science and Engineering, Johns Hopkins University, Baltimore, MD 21218, USA

**Keywords:** Parkinson’s disease, induced pluripotent stem cells, dopaminergic neuron, substantia nigra, autologous transplantation, allogeneic transplantation

## Abstract

Parkinson’s Disease (PD) is a neurodegenerative disease characterized by the progressive loss of dopaminergic neurons of the substantia nigra pars compacta with a reduction in dopamine concentration in the striatum. It is a substantial loss of dopaminergic neurons that is responsible for the classic triad of PD symptoms, i.e., resting tremor, muscular rigidity, and bradykinesia. Several current therapies for PD may only offer symptomatic relief and do not address the underlying neurodegeneration of PD. The recent developments in cellular reprogramming have enabled the development of previously unachievable cell therapies and patient-specific modeling of PD through Induced Pluripotent Stem Cells (iPSCs). iPSCs possess the inherent capacity for pluripotency, allowing for their directed differentiation into diverse cell lineages, such as dopaminergic neurons, thus offering a promising avenue for addressing the issue of neurodegeneration within the context of PD. This narrative review provides a comprehensive overview of the effects of dopamine on PD patients, illustrates the versatility of iPSCs and their regenerative abilities, and examines the benefits of using iPSC treatment for PD as opposed to current therapeutic measures. In means of providing a treatment approach that reinforces the long-term survival of the transplanted neurons, the review covers three supplementary avenues to reinforce the potential of iPSCs.

## 1. Introduction

Advances in medical treatments, surgical procedures, and cutting-edge technology have paved the way for groundbreaking discoveries into therapeutic development for formerly incurable diseases, notably neurodegenerative diseases. With countries sustaining an ever-growing aging population, the burden of neurodegenerative diseases is an especially pertinent health issue. Clinicians and researchers must now approach treatment strategies in a synergistic manner, working to translate novel scientific discoveries into effective, equitable therapeutics [1]. With the ability to target the root cause of disease and replenish unhealthy cellular populations, stem cell regenerative therapy provides an especially promising prognosis for PD.

### 1.1. Overview of Parkinson’s Disease

Parkinson’s Disease (PD) is the second most common neurodegenerative disorder, nearly affecting 2% of the population over the age of 65 [2]. PD is characterized by a progressive loss of dopaminergic neurons of the substantia nigra, causing a number of motor symptoms to arise, including tremors, rigidity, and bradykinesia, with postural instability appearing in patients as the disease progresses [3]. Another characteristic feature of PD is the existence of Lewy Bodies (LB) which are composed of the misfolded aggregates of α-synuclein (α-syn) protein. The neuron-to-neuron transmission of α-syn aggregates adds to the loss of dopaminergic neurons in the substantia nigra (SN). These dopaminergic neurons play a crucial role in facilitating inter-neuronal communication, as they are responsible for synthesizing and releasing dopamine, a neurotransmitter essential for brain communication and movement regulation. In the context of PD, there is a direct connection between the compromised functionality of dopaminergic neurons and their crucial role in transmitting signals to the basal ganglia, a region responsible for initiating and regulating movement patterns [4]. Oftentimes, patients will not develop symptoms and, therefore, not be diagnosed until 80 percent of dopamine neurons are lost. PD symptoms such as tremors, slowness of movement, stiffness, and balance problems occur [5]. Consequently, this disruption in dopamine transmission hinders the normal functioning of the basal ganglia, resulting in the impairment of movement coordination and control. Developing disease-modifying therapies to prevent neurodegeneration and preserve dopaminergic neurons is of paramount importance to improve the quality of life for PD patients [1,2].

### 1.2. Brief Overview of Current Treatment Options for PD

Existing strategies for managing PD are symptomatic and typically involve the replacement of DA neurotransmission by DA drugs, which relieve the patients of some of their motor symptoms [6]. Levodopa, also known as L-dopa, is a precursor of dopamine that crosses the blood–brain barrier and is converted into dopamine in the brain. This conversion helps to replenish the depleted dopamine levels in the brain, providing relief from motor symptoms such as rigidity, bradykinesia, and tremors. However, its long-term use may be associated with complications, such as motor fluctuations and dyskinesias, which may limit its effectiveness over time [7]. Alternatively, deep brain stimulation (DBS) is a surgical procedure that involves implanting electrodes into specific regions of the brain to deliver electrical impulses to modulate the activity of these brain regions [8]. As the disease progresses, these treatment approaches may lose efficacy and result in undesirable side effects, including cognitive dysfunction, further highlighting the need for PD treatments that address the disease’s underlying cause [9].

### 1.3. Levodopa

One of the most common pharmacological treatments for PD is Levodopa, a dopamine precursor in the form of a pill that has been widely used for over 40 years [10]. Levodopa is often combined with a dopa-decarboxylase inhibitor, inhibiting the synthesis of peripheral dopamine by aromatic amino-acid decarboxylase enzymes in the blood [10]. The benefits of pharmacological treatment fade with disease progression, and prolonged usage of these medications often results in side effects. After three to five years of treatment, motor fluctuations, loss of efficacy, and development of involuntary movements known as Levodopa-induced dyskinesia (LID) occur, significantly limiting the usefulness of L-DOPA as a therapeutic tool [11]. The administration of L-DOPA supplements helps alleviate motor symptoms but does not halt the progressive degeneration of DA neurons [12]. Approximately 49% of PD patients treated with levodopa developed dyskinesia after 5 years of treatment, and this percentage increases to 55% after 6 years of treatment [13]. While levodopa remains a valuable symptomatic treatment, complementary approaches, such as stem cell-based therapies and targeted interventions to inhibit α-syn aggregation, hold promise in addressing the neurodegenerative aspects of the disease and improving the overall therapeutic outcomes for PD patients.

### 1.4. Deep Brain Stimulation

Alternatively, surgical strategies, such as deep brain stimulation (DBS), have been shown to alleviate PD motor symptoms and offer symptomatic relief that cannot be controlled with medications [14,15]. However, DBS is limited to early- to-mid PD stages and may lose effectiveness after a few years. Although DBS surgery improves motor symptoms and quality of life, some symptoms may remain resistant to stimulation. Furthermore, the implantable pulse generator’s limited lifespan may lead to a potential worsening of Parkinsonian symptoms or even life-threatening crises upon withdrawal from the symptomatic benefits [16]. A study comparing the short and long-term effects of DBS on gait dysfunction in PD patients revealed adverse effects leading to decreased quality of life. Short-term stimulation (1–2 years) showed a positive impact on PD symptoms, whereas long-term stimulation (5+ years) demonstrated a significant decline in the beneficial effects of subthalamic nucleus DBS, particularly affecting gait and postural abilities [17]. While these treatments offer symptomatic relief, they do not address the root cause of the disease nor prevent the progressive degeneration of dopaminergic neurons in the substantia nigra. 

## 2. Cell Therapies for Parkinson’s Disease

Given the limitations of the current treatment, there is a growing focus on developing innovative therapies that target the root cause of PD. One promising avenue is cell therapy, which holds great potential in replenishing the lost dopaminergic neurons and restoring dopamine production in the brain [18]. Cell therapy involves the transplantation of healthy and functional cells, such as dopaminergic neurons, into the brain to replace the damaged neurons and potentially halt the neurodegenerative process [19]. This approach aims to provide a sustainable and long-lasting source of dopamine, offering a more comprehensive and targeted strategy to address the underlying cause of PD (Table 1). 

**Mesenchymal stem cells (MSCs):** MSCs have limited differentiation potential, typically giving rise to specific cell types such as bone, cartilage, and fat cells [20] (Figure 1). MSCs are obtained from donor sources, introducing compatibility and immune response challenges. MSCs, as adult stem cells, exhibit self-renewal and multilineage differentiation capabilities and can be isolated from various tissues, such as the umbilical cord, endometrial polyps, menses blood, bone marrow, and adipose tissue, making them practical for experimental and potential clinical use [21]. Reports have raised concerns about the potential tumorigenicity and tumor support or suppression effects of MSCs, necessitating long-term safety assessments. Ongoing clinical trials and comprehensive studies are essential to establish the safety profile of MSC-based therapies in human patients [22].

**Embryonic stem cells (ESC):** ESCs are generated from cells found in early human embryos known as blastocysts. ESCs possess the remarkable ability to differentiate into cell types representing all three germ layers– endoderm, mesoderm, and ectoderm– both in vitro and in vivo [23]. This unique property of ESCs holds tremendous potential for enhancing our understanding of early human embryology and advancing cell strategies for the treatment of various human diseases. However, the ethical concerns surrounding the destruction of human embryos are an imperative limitation of ESC-based clinical therapies [24]. In addition to the ethical concerns surrounding embryonic stem cells (ESCs), the clinical use of ESC-based therapy is impeded by safety issues. The inherent pluripotency of ESCs, which allows them to differentiate into various cell types, presents a challenge in terms of their control after transplantation [25]. When undifferentiated ESCs are transplanted, teratoma formation is likely characterized by developing tumors containing tissues from all three germ layers [26].

**Induced pluripotent stem cells (iPSC):** iPSCs are adult cells that have the ability to be reprogrammed or induced genetically to assume a stem cell-like state, allowing them to be further differentiated into a healthy cell type of interest [27]. The expression of the Yamanaka Factors—Oct4, Sox2, Klf4, and c-Myc—[28] enables the conversion of adult cells into iPSCs. The reprogramming process holds significant potential as a tool in gene editing procedures, allowing the insertion of regulatory genes into the cell’s genome [29]. iPSCs have gained attention as a promising pluripotent stem cell alternative with fewer ethical concerns, offering the ability to generate cells from all three germ layers and exhibit extensive proliferative capacity. However, iPSC’s clinical application is hindered by the risk of tumorigenicity, similar to ES cells [30]. An advantageous aspect of utilizing grafts derived from iPSCs compared to those derived from ESCs is the potential to generate autologous grafts allowing the patient’s own somatic cells to produce a neural grafting product, thereby obviating the need for immunosuppression, which would be necessary when using ESC-derived grafts [31]. iPSCs offer the advantage of being highly amenable to genetic manipulation for experimental purposes, as noted by Dr. R. Krolewski of Brigham and Women’s Hospital [32].

### 2.1. Transplantation Approach of iPSCs

The autologous transplantation of patient-specific iPSC-derived neurons is a potential clinical approach for the treatment of neurological disease. In autologous stem cell transplantation, the patient’s own cells are used for the transplant, while allogeneic transplants are obtained from a different individual, either a matched related or unrelated donor, posing the risk of immune rejection. Based on individualized risk factors, such as the unique set of genetics influencing the development of PD combined with environmental exposures, the ability to alter the genetic makeup of a patient provides another advantage of tailoring treatments [32]. Being able to identify the genes that make them susceptible to PD may give researchers clear therapeutic targets. Recent evidence has confirmed that neurons begin losing their normal functions and morphologies even before neuronal death, suggesting that simply preventing these neurons from dying is unlikely to be a practical therapeutic approach [33]. Hence, cell transplantation appears to be a potentially effective approach for the treatment of PD, as it may halt the progression of the disease by addressing the underlying cause of PD by replacing the depleted dopaminergic neurons.

### 2.2. Methods to Increasing Survival of Transplanted Neurons

The cell transplantation of iPSC-derived dopaminergic neurons alone is not a complete therapeutic approach due to the persisting neurotoxic microenvironment caused by α-syn aggregation. α-Syn, a protein found in Lewy bodies, has been closely associated with neurodegeneration, particularly in dopaminergic neurons [34]. The aggregates disrupt the functionality of neurons, impairing their ability to communicate effectively and ultimately triggering a cascade of neurotoxic events [35]. The selective vulnerability of these neurons to α-syn-induced damage was first observed in studies involving the neurotoxin 1-methyl-4-phenyl-1,2,3,6-tetrahydropyridine (MPTP), which resulted in the death of substantia nigra dopaminergic neurons [36]. The detrimental effects of α-syn aggregates extend beyond their presence in Lewy bodies. These aggregates may enter mitochondria and disrupt energy production, leading to mitochondrial dysfunction and increased oxidative stress [37]. Moreover, the presence of α-syn aggregates triggers inflammatory responses in the brain, activating immune cells like microglia and astrocytes, leading to the release of pro-inflammatory molecules [38]. Neuroinflammation exacerbates the neurotoxic environment and further contributes to the destruction of dopaminergic neurons. To achieve effective therapeutic outcomes, complementary approaches that address the underlying α-syn pathology and mitigate the neurotoxic environment are vital alongside cell transplantation (Figure 2).

### 2.3. Nanobodies

The ability of transplanted cells to be resilient against the impact of misfolded proteins originating from the host is necessary for long-term survival [39]. α-syn aggregation needs to be targeted because it may be released into extracellular spaces and taken up by adjacent cells, where they induce further misfolding and aggregation of protein [40].

Nanobodies are single-domain antibodies derived from camelid species that have emerged as a promising therapeutic approach for targeting α-syn aggregation and enhancing the survival of transplanted iPSC-derived dopaminergic neurons [41]. Antibodies 9E4, 1H7, and 5C1, which are nanobodies, were evaluated in a mouse model of PD to assess their impact on α-syn aggregation and motor function. Stereological analysis revealed a significant reduction in α-syn immunoreactivity in the neuropil of animals treated with these nanobodies compared to the control group. Additionally, immunohistochemical analysis with the SYN105 antibody demonstrated a significant reduction in abnormal α-syn aggregates in both the temporal cortex and striatum of the treated mice. Notably, treatment with antibody 5D12, another nanobody, resulted in a significant reduction in α-syn aggregates in the neocortex but not in the striatum, indicating the specificity of nanobodies in targeting α-syn aggregates in different brain regions. These findings highlight the potential of nanobodies as targeted immunotherapies to combat α-syn aggregation in PD, addressing region-specific neurotoxicity and contributing to the preservation of dopaminergic neurons [42]. The nanobody PFFNB2 can specifically recognize α-syn PFF over α-syn monomers to significantly dissociate α-syn fibrils. This strategy may offer protection to the transplanted neurons, potentially reducing their vulnerability to neurodegeneration and promoting their successful integration and function within the host brain [41].

### 2.4. LAG3-Related Pathway

Lymphatic activation gene 3 (LAG3) is a receptor that facilitate the pathogenic α-syn and tau cell-to-cell transmission [43,44,45]. Genetic depletion of LAG3 and anti-LAG3 antibodies can significantly block pathogenic α-syn aggregation, inflammation, neurotoxicity in vitro and in vivo as well as other prion-like seeds [46,47]. Moreover, biochemical and behavioral deficits associated with PD pathology are also mitigated, indicating the potential of LAG3-targeted therapies in preventing pathology propagation, preserving dopaminergic function and improving motor outcomes. By transplanting LAG3 deficient neurons or using LAG3 antibodies, the internalization of α-syn by transplanted neurons may be decreased, thereby likely leading to their reduced vulnerability to α-syn-induced neurotoxicity [47]. This attenuation of α-syn transmission to the transplanted cells not only enhances their survival but also contributes to the overall preservation of the neuronal network and motor function in PD [46]. Furthermore, the use of LAG3-targeted therapies offers a potential means of addressing region-specific neurotoxicity, as the localization and role of LAG3 in propagating α-syn pathology may vary across different brain regions [43].

### 2.5. Exosomes

Exosomes are nano-sized vesicles released by various cell types, including neurons, astrocytes, and microglia, in the brain. These extracellular vesicles have garnered interest due to their ability to shuttle genetic material and proteins between cells, thereby influencing gene expression and protein activity in recipient cells [48]. Exosomes isolated from the blood in an NHP model were loaded with a saturated dopamine solution, serving as potential carriers for targeted therapy in PD. The interaction between transferrin and the transferrin receptor allowed the dopamine-loaded exosomes to efficiently cross the blood–brain barrier, resulting in enhanced therapeutic efficacy and reduced toxicity compared to systemic intravenous delivery of free dopamine [49].

Exosomes have shown promise in targeting neuroinflammation, a critical aspect of PD pathology. In a study, exosomes loaded with catalase, an antioxidant enzyme, were investigated for their potential to alleviate neural inflammation and enhance neuronal survival in a PD mouse model. To evaluate the therapeutic effects, C57BL/6 mice were stereotactically injected with 6-OHDA into the substantia nigra pars compacta to induce PD-like pathology. The mice were then intranasally injected with exosomes containing catalase activity. Two exoCAT formulations were evaluated: catalase-loaded exosomes obtained by sonication and permeabilization with saponin. Control groups included PD mice injected with phosphate-buffered saline (PBS) and healthy mice injected with PBS or empty exosomes. Both exoCAT formulations significantly reduced brain inflammation and promoted neuronal survival in the PD mouse model. The encapsulation of catalase into exosomes enhanced its therapeutic efficacy, potentially by preserving enzymatic activity, prolonging circulation time, reducing immunogenicity, and improving interactions with epithelial cells for enhanced drug transport. exoCAT obtained by permeabilization with saponin exhibited superior therapeutic effects compared to those obtained by sonication, possibly due to better surface morphology and reduced clearance by macrophages [50].

Exosomes containing modified α-syn siRNAs reduced the amount of α-syn mRNA transcription and translation in the brain of transgenic mice [51]. Moreover, exosomal shRNA minicircles to target α-syn in a PD mouse model resulted in reduced α-syn aggregation, decreased dopaminergic neuron death, and improved clinical symptoms [52]. Thus, exosomes present a promising avenue for delivering therapeutic agents or genetic modulators to attenuate neuroinflammation in PD, which may complement the cell therapy approach with iPSC-derived dopaminergic neurons.

## 3. Animal and Human Models

### 3.1. Nonhuman Primates (NHPs)

iPSCs have been proven to potentially impact many areas of medicine and therapeutics, highlighting the potential of the instrument in various fields of biomedicine. iPSCs have become a powerful tool in basic, as well as translational, and clinical research because of their ability to be maintained indefinitely while preserving the host’s genetic makeup. This method was tested on a nonhuman primates (NHPs) model in which autologous DA cells were introduced into the brain of a cynomolgus monkey PD model without immunosuppression; three PD monkeys that had received no grafts served as controls. The PD monkey that had received autologous grafts experienced behavioral improvement compared with that of controls [53]. In addition, emerging regenerative medicine therapies are being developed using neurons derived from autologous stem cells, enabling the development of patient-specific treatments [54]. Furthermore, technological advancements that enable genetic manipulation of iPSCs have ensured they have become an option for cell therapy. The successful clinical application of iPSC-derived midbrain-like dopamine neurons in treating PD necessitates the careful consideration of factors such as the survival of a minimum required dose of transplanted neurons and the sufficient reinnervation of the denervated putamen. These factors are crucial for achieving improvements in PD motor symptoms [55,56]. In essence, the transplantation of iPSC-derived dopaminergic neurons is a potential area of focus for the treatment of PD, as there must be sufficient levels of dopamine in the brain to function accordingly.

To enhance preclinical studies and improve predictive validity for clinical use, it is essential to utilize animal models that closely mimic PD as observed in patients [57]. NHPs have been extensively employed in research and nonclinical development over the past decades due to their genetic, anatomical, physiological, and immunological similarities to humans [58]. A study found the protocol of NCAM(+)/CD29(low) sorting to result in enriching ventral midbrain dopaminergic neurons from the pluripotent stem cell-derived neural cell populations. Further, these neurons also exhibited increased expression of FOXA2, LMX1A, TH, GIRK2, PITX3, EN1, and NURR1 mRNA. These neurons were also found to bear the potential to restore motor function among the 6-hydroxydopamine lesioned rats 16 weeks after transplantation [59]. Furthermore, the primate iPSC-derived neural cell was found to have survived without any immunosuppression after one year of autologous transplant, highlighting the proof-of-concept around the feasibility and safety of iPSC-derived transplantation for PD. Transplantation of iPSC-derived mDA neurons into rodent PD models robustly restores motor function and reinnervates the host brain while showing no evidence of tumor formation or redistribution of the implanted cells. The dopamine-producing neurons derived from iPSCs have improved behavior in a rat model of PD. As previously stated, in one patient, dopamine cells from autologous iPSCs seemed to stabilize or even slightly improve motor symptoms 18–24 months after transplantation. 

### 3.2. N-of-1 Report

In relation to the causes of PD, cell transplantation in patients with PD replaces the lost dopaminergic neurons of the substantia nigra pars compacta. Through this transplantation, the loss of dopaminergic neurons, responsible for the symptomatic motor deficits of PD, may be restored. Through an N-of-1 report held by Mass Brigham General Hospital, patient-derived midbrain dopaminergic progenitor cells were implanted into the putamen (left hemisphere followed by the right hemisphere, six months apart) of a patient with PD without the need for immunosuppression. Clinical measures of PD symptoms stabilized or improved at 18 to 24 months after implantation [29]. In this study, several clinicians and scientists took a patient’s cells, made them into iPSCs, and then converted them into dopamine neurons, grew them up in sufficient quantities to ensure they were safe, and put them back into that patient. As a result, the patient required less prescription of Levodopa, indicating the ability of iPSCs to control some of the PD symptoms and halt the progression of the disease [32]. With future applications of iPSC-based therapies, it is necessary to understand that iPSCs need to go through various quality checks to fully ensure the reprogramming and differentiation factors were successful and check the development of any dangerous cell growths, most known as teratomas.

## 4. Research Methods and Data Collection

Due to the preliminary evidence suggesting the overall benefits of iPSC replacement therapy for PD, a literature search was conducted to evaluate the effectiveness of iPSCs as a viable treatment option for PD patients as they may restore the lost dopaminergic neurons and improve motor coordination. Four research articles were analyzed that used the Movement Analysis Panel (MAP) to test separable components of arm motion and simple and complex finger coordination. Data on NHPs were primarily used due to the limited research on a human test subject worldwide using iPSCs-derived dopaminergic neurons as a potential treatment option. The studies implemented various methods, including autologous and allogeneic transplantation approaches, behavioral assessments, neuroimaging techniques, and histological analyses, and are presented in detail in Table 2.

## 5. Results and Discussion

Through clinical applications, the autologous transplantation approach utilizes iPSCs derived from patients with PD to generate dopaminergic neurons. These dopaminergic neurons may be successfully transplanted into the adult rodent striatum without signs of neurodegeneration [60].

In preclinical studies, autologous transplantation of iPSC-derived dopaminergic neurons has been suggested to be a promise. In a non-human primate (NHP) model, iPSC-derived dopaminergic neurons were transplanted, leading to functional recovery and the survival of approximately 20,000 tyrosine hydroxylase (TH)-positive neurons in the graft [60]. The feasibility of iPSC-based transplantation in restoring dopaminergic function was demonstrated through this approach. Moreover, autologous transplantation of iPSC-derived dopaminergic neurons in cynomolgus monkeys with induced parkinsonism resulted in sustained motor improvements. The transplanted animals exhibited increased daytime activity counts, indicating functional recovery that persisted for up to 18 months (146% increase compared to pre-transplantation activity levels) as analyzed by the Movement Analysis Panel. PET neuroimaging further confirmed increased dopamine reuptake in the transplanted putamen, providing additional evidence of the effectiveness of iPSC-derived dopamine neurons in restoring dopamine levels [56].

Allogeneic transplantation of iPSC-derived dopaminergic neurons has also been explored as an alternative approach. In a rat model, the transplantation of PD patient iPSC (PDiPS) cell-derived dopaminergic neurons demonstrated a significant reduction in contralateral rotations and improved motor function [61]. To assess the long-term effects of iPSC-derived dopaminergic neurons, studies have investigated the survival and integration of the transplanted cells. Autologous transplantation in NHP models revealed robust survival of TH-positive neurons for up to 1.5 years, accompanied by behavioral improvements and increased binding sites of the dopamine transporter [60]. These findings support the long-term viability and functional benefits of iPSC-derived dopaminergic neurons. Furthermore, iPSC-derived dopaminergic neurons demonstrated safety and regenerative abilities, as autologous transplantation demonstrated remarkable reinnervation of the denervated putamen without the need for immunosuppression. Notably, no graft overgrowth, tumor formation, or inflammatory reactions were observed [56,60].

Collectively, these studies propose the use of iPSC-derived dopaminergic neurons for transplantation as a potential therapeutic approach for PD. The transplantation of iPSCs into dopaminergic neurons demonstrates their feasibility and the absence of neurodegeneration upon transplantation into the adult rodent striatum [60]. Animal studies, including NHP and rat models, have provided evidence of improvement and an increase in dopaminergic neurons following both autologous and allogeneic transplantation of iPSC-derived dopaminergic neurons [56,61]. These studies also highlight the survival of transplanted cells, long-term behavioral improvements (including a 146% increase in daytime activity counts), and functional effects mediated by iPSC-derived dopaminergic neurons. Additionally, the safety and regenerative abilities of iPSC-derived dopaminergic neurons have been demonstrated, with no observed complications such as graft overgrowth, tumor formation, or inflammatory reactions. These findings support the potential of iPSC-based treatments for PD, underscoring their therapeutic potential in restoring dopamine levels and alleviating motor symptoms associated with the disease.

## 6. Limitations

Despite the promising results of iPSC-derived dopaminergic neuron transplantation, there are several limitations and challenges that need to be addressed. Successful transplantation requires optimizing differentiation protocols and improving the survival of midbrain dopamine neurons [56]. Consistency in protocols and the refinement of clinical procedures are necessary for future clinical trials [56]. Additionally, the potential risk of tumor formation and the need for alternative reprogramming methods to avoid genetic modifications are areas that require further research and development [62].

The main limitations of the current research on iPSC-derived dopaminergic neurons and their transplantation are primarily associated with the small-scale studies conducted and the use of NHP models. The small sample size in the conducted studies is a significant limitation, as it may not fully represent the diversity and heterogeneity of PD patients, making it challenging to draw definitive conclusions about the efficacy and safety of iPSC-based therapies. Moreover, the use of NHPs as experimental subjects introduces potential differences between animal models and humans, including genetic variations, immune responses, and anatomical disparities, which may influence the outcomes observed in NHP studies and hinder accurate extrapolation to human patients. In addition to the regulatory challenges, the sourcing of iPSCs and their subsequent differentiation into dopaminergic neurons highlight the potential for tumorigenicity, which must be carefully addressed before these therapies may be widely implemented. Future research endeavors should focus on conducting large-scale clinical trials, exploring alternative animal models, and addressing ethical and regulatory considerations to overcome these limitations associated with iPSC-based therapies.

The process of addressing all patients with iPSC-based therapies for PD poses challenges due to its cumbersome nature and associated expenses. Treating many patients with personalized iPSC-derived dopaminergic neurons requires substantial resources in terms of equipment, personnel, and materials. The extensive laboratory work involved in reprogramming iPSCs into dopaminergic neurons adds to the complexity and cost of the process. During the reprogramming process, it is crucial to ensure the absence of mutations that could impact the functionality and safety of the derived neurons. The genetic sequencing of iPSCs used for transplantation is necessary to verify their genetic integrity. Complete sequencing allows the identification of potential mutations that may compromise the quality and effectiveness of the derived dopaminergic neurons. By performing comprehensive genetic sequencing, researchers may ensure that the iPSC-derived dopaminergic neurons are free from harmful genetic alterations, minimizing the risk of adverse effects in patients. One specific concern during reprogramming is the avoidance of oncogenic mutations, which are genetic alterations that may lead to tumor formation or cancer. To ensure the safety of iPSC-derived dopaminergic neurons, stringent quality control measures, such as rigorous genetic screening and validation, are employed to identify and exclude iPSCs carrying oncogenic mutations. These measures ensure that the resulting dopaminergic neurons are free from genetic abnormalities that could potentially lead to tumor formation. The process of addressing all patients with iPSC-based therapies for PD is costly and resource-intensive.

Lastly, the effectiveness of iPSC-based therapies for PD is likely to be influenced by the disease stage at which the treatment is administered. In early-stage PD, when α-syn aggregation and neuronal damage are relatively localized, iPSCs present a more favorable opportunity for restoring function and improving symptoms. The transplanted iPSC-derived neurons have a better chance of integrating into the affected brain regions and replenishing the depleted dopaminergic neurons. However, in the later stages of the disease, when α-syn aggregates have spread extensively throughout multiple brain regions, iPSCs may encounter greater challenges in their therapeutic potential. Thus, it is essential to consider the disease stage when developing iPSC-based treatments and explore complementary approaches with nanobodies for addressing the more advanced neurodegenerative changes in PD.

## 7. Conclusions

The studies reviewed in this analysis provide a foundation for the future application of iPSCs in the treatment of PD. They demonstrate the potential of iPSC-derived dopaminergic neurons as a viable option for cell therapy in PD patients. To overcome the limitations of the current research, further studies on a larger scale and with human participants are necessary. Conducting rigorous clinical trials to establish the safety, efficacy, and long-term outcomes of iPSC-based therapies in PD patients is crucial. Additionally, comparative studies between NHP models and human subjects could help address potential discrepancies and improve the translatability of the findings.

The integration of cell-based therapy, specifically through the transplantation of iPSC-derived dopaminergic neurons, combined with the strategic use of nanobodies, LAG3, and exosomes targeting α-syn aggregate spreading and neurotoxicity, represents a promising and ideal approach for treating PD. As the burden of neurodegenerative diseases continues to rise, finding innovative and effective treatment options becomes imperative. Neuron survival is crucial for maintaining motor and cognitive function, and cell therapy has shown significant potential in preclinical studies. iPSCs offer a unique advantage in treating PD, as they may renew healthy cell populations in targeted locations where neuronal death is relatively localized. This targeted and patient-specific approach allows for precision in combating degeneration while minimizing invasiveness. Through ongoing research and careful clinical implementation, cell therapy and the innovative use of targeted therapies may pave the way for a brighter future for those living with PD.

## Figures and Tables

**Figure 1 pharmaceutics-15-02656-f001:**
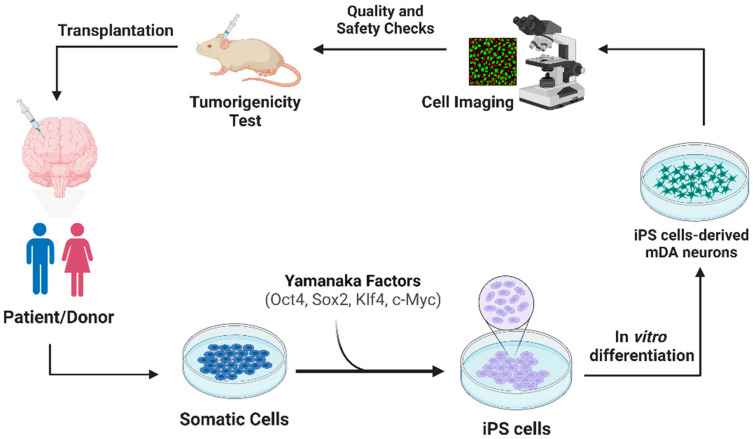
iPSC-Based Cell Therapy for Parkinson’s Disease. Description: Patient-derived somatic cells are reprogrammed into iPSCs using Yamanaka factors, differentiated into iPSC-derived mDA neurons, rigorously imaged for accuracy, evaluated in a non-human primate (NHP) model for tumorigenic potential, and ultimately transplanted back into the patient.

**Figure 2 pharmaceutics-15-02656-f002:**
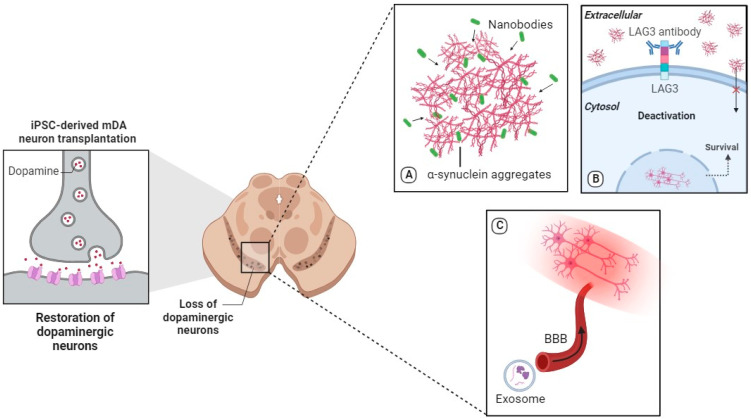
Cell Therapy for PD. Description: Transplantation of iPSC-derived dopaminergic neurons causes increased dopamine reuptake and reinnervation of the transplanted putamen by iPSC-derived dopamine neurons. (**A**), nanobodies target neurotoxicity by neutralizing harmful neurotoxic proteins and enhancing neuronal survival. (**B**), LAG3-directed antibodies target α-syn aggregate spreading by inhibiting the internalization and propagation of misfolded α-syn between neurons. (**C**), exosomes release anti-inflammatory factors to target neuroinflammation, promoting a neuroprotective microenvironment and supporting the survival of transplanted dopaminergic neurons.

**Table 1 pharmaceutics-15-02656-t001:** Comparison of different properties for cell therapy.

	MSCs	ESCs	iPSCs
Differentiation Potential	✓ *	✓	✓
Allogeneic and Autologous Source Availability	✓	✕	✓
Ethical Concerns	✓	✕	✓
Immunological Compatibility	✓	✕	✓
Restoration of Brain Function on its Own	✕	✕	✓
Tumorigenicity	✓	✕	✕
Genetic Manipulation	✕	✓	✓

* ✓ safe/advantage; ✕ concern/disadvantage.

**Table 2 pharmaceutics-15-02656-t002:** Summary of current applications of iPSCs for treating PD.

Cell Sources	Subjects	Effectiveness	Limitations	Publications
DA neurons derived from directly reprogrammed fibroblasts	Parkinsonian NHPs	-Robust survival of TH+ neurons (∼20,000) at 1.5 years post-transplantation -Long-term functional recovery of PD-like motor symptoms -Increased DA transporter binding sites using PET neuroimaging -Survival of >13,000 engrafted DA neurons	-Study conducted on non-human primates (NHPs), need for further assessment in human subjects -Limited information on long-term effects beyond 1.5 years post-transplantation	[60]
CM-iPSC-derived midbrain dopamine neurons	MPTP-lesioned cynomolgus monkeys (CMs)	-Gradual functional motor improvement -Increased daytime activity counts -Improved limb motor function -PET neuroimaging confirmed increased dopamine reuptake -13,029 surviving transplanted dopaminergic neurons in the putamen -No immunosuppression	-Long-term function, survival, and safety of iPSC-derived dopamine neurons in non-human primate PD model not established. -Need for consistent protocols to enhance midbrain dopamine neuron survival and achieve functional improvements. -Further research is needed to optimize dopamine neuron differentiation and refine clinical protocols	[56]
PD patient iPSC-derived DA neurons	Adult striatum of unlesioned rats	-High number of neurons in all grafts; minimal presence of astroglial and microglial cells. -Transplanted rats showed a progressive reduction in amphetamine-induced rotations and contralateral rotations compared to the control group. -Successful integration of engrafted DA neurons into the host striatum; 4890 ± 640 dopamine (DA) neurons in the grafts. -No tumor formation was observed in grafts up to 16 weeks after transplantation.	-Limited connectivity observed between engrafted DA neurons and host striatum. -Insufficient improvements in cylinder and adjustment stepping tests indicate the need for better functional outcomes. -Inefficient differentiation protocol may hinder the precise patterning of iPSCs into specific A9 dopaminergic neuronal phenotypes. -Absence of Oct4 expression in factor-carrying PDiPS cells at 16 weeks suggests successful gene silencing.	[61]
Reprogrammed fibroblasts	6-OHDA-lesioned rats	-Marked recovery of rotation behavior 4 weeks after transplantation -High number of TH+ neurons in grafts and increased over time in culture -Proliferation of transplanted cells observed in graft areas. -Teratoma formations were detected initially but eliminated in transplanted rats	-Rats treated with 6-OH-DA may not fully represent PD in humans. -Retroviral vectors used in cell reprogramming may randomly integrate into the genome, potentially causing genetic changes or cancerous transformation. -16 out of 36 iPSC chimeras developed tumors, highlighting the need for improved safety measures.	[62]

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
