# Peer review of "Cell Therapy for Parkinson’s Disease"

_pharmaceutics, 2023, doi:10.3390/pharmaceutics15122656_

Round 1

Reviewer 1 Report

Comments and Suggestions for Authors

The manuscript examines new methods that may be used in the future to treat Parkinson's disease. A review of the effects of dopamine on patients with Parkinson's disease is presented, various types of treatment using stem cells are described, and a conclusion is drawn about the advantage of Induced Pluripotent Stem Cells over MSCs and ESCs in the treatment of Parkinson's disease.

The authors specify that future transplantation studies of dopaminergic neurons derived from Induced Pluripotent Stem Cells should combine therapy by adding nanobodies, antibodies targeting lymphocyte activation gene 3 (LAG3), and exosomes in patients with Parkinson's disease.

There are several comments to the article:

There are no references to literary sources in the Cell Therapies for Parkinson’s Disease section.

The statement on lines 100-101 also requires a reference to the literary source: “When undifferentiated ESCs are transplanted, teratoma formation is likely characterized by developing tumors containing tissues from all three germ layers”

The same for lines 111-112: “However, iPSC’s clinical application is hindered by the risk of tumorigenicity, similar to ES cells.”

Author Response

Thank you very much for taking the time to review this manuscript. Please find the detailed responses below and the corresponding corrections highlighted in the re-submitted file.

1. There are no references to literary sources in the Cell Therapies for Parkinson’s Disease section.

Thank you for the comment. We cited two literary sources in the text. Line 107, “One promising avenue is cell therapy, which holds great potential in replenishing the lost dopaminergic neurons and restoring dopamine production in the brain”. We cited the reference Qian et al., Nature, 2020 (No.18, line 108).

Line 108, “Cell therapy involves the transplantation of healthy and functional cells, such as dopaminergic neurons, into the brain to replace the damaged neurons and potentially halt the neurodegenerative process”. We cited the reference, Wu et al., 2010, Molecules (No. 19, line 110).

2. The statement on lines 100-101 also requires a reference to the literary source: “When undifferentiated ESCs are transplanted, teratoma formation is likely characterized by developing tumors containing tissues from all three germ layers”.

Thank you for the comment. In the text, line 131, “When undifferentiated ESCs are transplanted, teratoma formation is likely characterized by developing tumors containing tissues from all three germ layers”. In this statement, we cited the reference, Hentze et al., Stem Cell Res, 2009 (No. 26, line 132).

3. The same for lines 111-112: “However, iPSC’s clinical application is hindered by the risk of tumorigenicity, similar to ES cells.”

Thank you for the comment. In the text, line 140, “However, iPSC’s clinical application is hindered by the risk of tumorigenicity, similar to ES cells”. We cited the reference, Lee et al., Nat Med, 2013 (No. 30, line 141).

Thank you for the comments, and hopefully, our modifications could meet your requirements.

Reviewer 2 Report

Comments and Suggestions for Authors

Shastry and colleagues presented a review entitled “Cell Therapy for Parkinson’s disease" that describes the potential use of the cell therapy for the treatment of Parkinson’s disease, showing both the benefits and limitations in using this therapy. The review deals with a topic of high significance, adding new information to the reviews already present dealing with this topic, however some important modifications should be done to improve the manuscript.

1.     Many references are present within the text but not in the references section. Please include all the articles in the references section.

2.     Line 55, section “Brief Overview of Current treatment options for PD”: bibliography referring to the use and limitations of L-Dopa is completely missing.

3.     The structure of the review could be better arranged, I suggest moving the sections “Levodopa” line 134 and “Deep brain stimulation” line 152 immediately after the section “Brief Overview of Current treatment options for PD” and before the section on the cell therapies.

4.   The section “Advantage of iPSCs” line 129 does not report any information regarding the benefits of using iPSCs, I suggest expanding the section or removing it.

5.     Line 226-227. The authors state:” The neutralization of α-synuclein aggregation by nanobodies offers protection to the transplanted neurons, reducing their vulnerability to neurodegeneration and promoting their successful integration and function within the host brain”. Is this statement supported by experimental data or is it an assumption of the authors? If yes, add the reference, otherwise rephrase the sentence.

6.     Line 245-247, sentences “In addition to its role in halting α-synuclein aggregation, targeting LAG3 represents a promising strategy to protect transplanted dopaminergic neurons from neurodegeneration and promote their successful integration within the host brain (Zhang el al., 2021)”. The information reported by the authors in this sentence is not in the paper cited, it seems to be an assumption of the authors and not an experimental evidence reported in the paper. Please provide the correct reference or rephrase it.

7.     Table 1. The reference Osborn et al 2020 is referred to “Osborn, T. M., Hallett, P. J., Schumacher, J. M., & Isacson, O. (2020). Advantages and recent developments of 609 autologous cell therapy for parkinson's disease patients. Frontiers in Cellular Neuroscience, 14. 610 https://doi.org/10.3389/fncel.2020.00058” that is a review, please provide the research article to which the study refers.

Author Response

Thank you for taking the time to review this manuscript. Please find the detailed responses below and the corresponding revisions highlighted in the re-submitted file.

1. Many references are present within the text but not in the references section. Please include all the articles in the references section.

Thank you for pointing out this issue. We updated the references to align with the in-text citations.

In the text, line 65, “Existing strategies for managing PD are symptomatic and typically involve the replacement of DA neurotransmission by DA drugs, which relieve the patients of some of their motor symptoms”. We cited the paper, Zohoor et al., Parkinson’s disease, 2018, in the references section (NO. 6, line 436).

In the text, line 69, “However, its long-term use may be associated with complications, such as motor fluctuations and dyskinesias, which may limit its effectiveness over time”. We cited the paper, Stacy, M et al., Clin Neuropharmacol, 2008, in the references section (No. 7, line 439).

In the text, line 107, “One promising avenue is cell therapy, which holds great potential in replenishing the lost dopaminergic neurons and restoring dopamine production in the brain”. We cited the paper, Qian, H., et al., Nature, 2020, in the references section (No. 18, line 466).

In the text, line 108, “Cell therapy involves the transplantation of healthy and functional cells, such as dopaminergic neurons, into the brain to replace the damaged neurons and potentially halt the neurodegenerative process”. We cited the paper, Wu, Y.-P., et al., Molecules, 2010, in the references section (No. 19, line 468).

In the text, line 131, “When undifferentiated ESCs are transplanted, teratoma formation is likely characterized by developing tumors containing tissues from all three germ layers”. we cited the paper, Hentze et al., Stem Cell Res, 2009, in the references section (No. 26, line 483).

In the text, line 140, “However, iPSC’s clinical application is hindered by the risk of tumorigenicity, similar to ES cells”. We cited the paper, Lee et al., Nat Med, 2013, in the references section (No. 30, line 491).

In the text, line 199, “The nanobody, PFFNB2, can specifically recognize α-syn PFF over α-syn monomers to significantly dissociate α-syn fibrils. This offers protection to the transplanted neurons, potentially reducing their vulnerability to neurodegeneration and promoting their successful integration and function within the host brain”. We cited the paper, Butler, Y., et al.,Nat Commun, 2022, in the references section (No. 43, line 518).

In the text, line 204, “Lymphatic activation gene 3 (LAG3) is a receptor that facilitate the pathogenic α-syn and tau cell-to-cell transmission”. We cited the papers, Zhang, S., et al, Proc Natl Acad Sci USA, 2021; Mao, X., et al., bioRxiv, 2021; Chen, C., et al., bioRxiv, 2023 in the references section (No. 44-46, line 520, 523, 525).

In the text, line 205, “Genetic depletion of LAG3 and anti-LAG3 antibodies can significantly block pathogenic α-syn aggregation, inflammation, neurotoxicity in vitro and in vivo as well as other prion-like seeds”. We cited the papers, Gu, H., et al., Front Cell Neurosci, 2021; Mao, X., et al., Science, 2016, in the references section (No. 47-48, line 527, 529).

In the text, line 312, “DA neurons derived from directly reprogrammed fibroblasts”. We cited the paper, Osborn T., D.D., Stem Journal, 2020, in the references section (No. 61, line 560).

2. Line 55, section “Brief Overview of Current Treatment Options for PD”: bibliography referring to the use and limitations of L-Dopa is completely missing.

Thank you for the comment. We cited the two literary sources in the Brief Overview of Current Treatment Options for PD section.

In the text, line 65, “Existing strategies for managing PD are symptomatic and typically involve the replacement of DA neurotransmission by DA drugs, which relieve the patients of some of their motor symptoms”. We cited the paper, Zohoor et al., Parkinson’s disease, 2018, in the references section (NO. 6, line 436).

In the text, line 69, “However, its long-term use may be associated with complications, such as motor fluctuations and dyskinesias, which may limit its effectiveness over time”. We cited the paper, Stacy, M et al., Clin Neuropharmacol, 2008, in the references section (No. 7, line 439).

3. The structure of the review could be better arranged, I suggest moving the sections “Levodopa” line 134 and “Deep brain stimulation” line 152 immediately after the section “Brief Overview of Current treatment options for PD” and before the section on the cell therapies.

Thank you for the comments. The treatments of Levodopa and Deep brain stimulation have been moved in the Brief Overview of Current Treatment Options for PD section to expand on the details of the earlier parts, in the line 76 and line 91. 

4. The section “Advantage of iPSCs” line 129 does not report any information regarding the benefits of using iPSCs, I suggest expanding the section or removing it.

Thank you for the comments. We removed the “Advantage of iPSCs” section and replaced it with “Transplantation Approach of iPSCs” (line 153).

5. Line 226-227. The authors state:” The neutralization of α-synuclein aggregation by nanobodies offers protection to the transplanted neurons, reducing their vulnerability to neurodegeneration and promoting their successful integration and function within the host brain”. Is this statement supported by experimental data or is it an assumption of the authors? If yes, add the reference, otherwise rephrase the sentence. 

Thank you for the comment. We revised the sentence (line 199) into “The nanobody, PFFNB2, can specifically recognize α-syn PFF over α-syn monomers to significantly dissociate α-syn fibrils”. We cited the paper, Butler, Y., et al.,Nat Commun, 2022 (No. 43, line 202). This is an assumption that using nanobody to dissolve the α-syn fibrils can improve the cell microenvironment for the transplanted neurons.   

6. Line 245-247, sentences “In addition to its role in halting α-synuclein aggregation, targeting LAG3 represents a promising strategy to protect transplanted dopaminergic neurons from neurodegeneration and promote their successful integration within the host brain (Zhang el al., 2021)”. The information reported by the authors in this sentence is not in the paper cited, it seems to be an assumption of the authors and not an experimental evidence reported in the paper. Please provide the correct reference or rephrase it.

Thank you for pointing out this issue. In the text, line 209, “By transplanting LAG3 deficient neurons or LAG3 antibodies, the internalization of α-syn by transplanted neurons may be decreased, thereby reducing their vulnerability to α-syn-induced neurotoxicity”. We cited the reference Mao, X., et al., Science, 2016 (No. 48, line 211). The antibodies to LAG3 could reduce the pathology set in motion by the transmission of pathologic α-synuclein, which could be possible strategy to protect transplanted dopaminergic neurons.

7. Table 1. The reference Osborn et al 2020 is referred to “Osborn, T. M., Hallett, P. J., Schumacher, J. M., & Isacson, O. (2020). Advantages and recent developments of 609 autologous cell therapy for parkinson's disease patients. Frontiers in Cellular Neuroscience, 14. 610 https://doi.org/10.3389/fncel.2020.00058” that is a review, please provide the research article to which the study refers.

In the text, line 311, “DA neurons derived from directly reprogrammed fibroblasts”. We cited the paper, Osborn T., D.D., Stem Journal, 2020, in the references section (No. 61, line 311).

Thank you for the comments, and hopefully our modifications could meet your requirements.

Reviewer 3 Report

Comments and Suggestions for Authors

This article discusses the potential use of Induced Pluripotent Stem Cells (iPSCs) for the treatment of Parkinson's Disease (PD). PD is characterized by the loss of dopaminergic neurons, which leads to symptoms such as tremors and rigidity. Current therapies only offer symptomatic relief and do not address the underlying neurodegeneration. iPSCs can be used to create patient-specific models of PD and can be differentiated into dopaminergic neurons, offering a promising avenue for addressing neurodegeneration. The article discusses the benefits of iPSC treatment compared to current therapies and suggests three supplementary avenues to reinforce the potential of iPSCs for long-term survival of transplanted neurons. These include the addition of nanobodies, LAG3-directed antibodies, and exosomes in PD patients. Authors conclude that further research is needed to explore the potential of iPSC-derived dopaminergic neuron transplantation for PD patients. I consider it as a valuable article suitable for publiation in its present form.

Comments on the Quality of English Language

English quality is fine

Author Response

Thank you very much for the comments.

Round 2

Reviewer 2 Report

Comments and Suggestions for Authors

Thank you for revising the work according to my suggestions. I sustain that it is suitable for publication.